# HOW FAR CAN WE GO WITHOUT FINETUNING?

## ABSTRACT

Many of the existing deep learning methods are trained for scenarios which: (1) use (costly) fine-tuning of the latent spaces on the target dataset in the "downstream task" (2) do not account for continual and open-set learning (3) do not provide interpretability. Instead of trying to solve the problem of semi- and unsupervised learning through representation learning, we propose recasting it into the problem of analysing existing foundational models' feature spaces. We show that a simple baseline, based on non-parametric clustering analysis of the latent feature spaces and pre-trained classifiers on large-scale datasets, can help solve a set of such problems even without finetuning. It can also be seen as a set of metrics for assessment of generalisation within the latent feature spaces. We argue that better generalising pre-trained architectures can solve a number of problems without finetuning, providing the basis for lifelong learning without catastrophic forgetting and with a means of interpretation.

## 1 INTRODUCTION

Development of new methods for machine learning is often based on large datasets trained offline, often with supervised labels. To make it work for semi- and unsupervised learning scenarios, many works use the representation learning techniques involving finetuning of the feature space, Rao et al. (2019). However, this approach has a number of limitations:

1. it does not explicitly take into account the continually evolving data stream cases
2. such finetuning does not provide multi-task generalisation of already existing features
3. this approach does not account for interpretability

We show that one could build upon the latent spaces of classifiers, such as SWAG-ViT (Dosovitskiy et al. (2021); Singh et al. (2022)), trained on large amounts of data, to handle continually evolving data streams in an unsupervised or semi-supervised way, while providing interpretability through prototypes. The proposed approach constitutes a simple baseline showing competitive performance against existing continual learning methods. The proposed analysis suggests that with a generic enough feature space one could perform continual learning using only shallow learning techniques, such as clustering, within the pre-trained latent spaces without finetuning.

The list of the contributions of the paper include:

- We demonstrate that without any finetuning, clustering of latent representations of foundation models, such as ViT, can competitively solve a range of well-known problems in a unified framework: unsupervised clustering, few-shot learning, unsupervised domain adaptation and continual learning
- The proposed set of evaluations can serve as a simple baseline for quantifying generalisation without finetuning of the vision transformer models to a range of downstream tasks

Therefore, we argue that with the advancement of foundation models, improvements in semi- and unsupervised learning methods can shift from end-to-end training toward clustering and decision making over the foundation models' latent space. One may also suggest that the proposed framework can help solve other tasks such as reinforcement and collaborative learning. Furthermore, such scheme, through its simplicity, provides interpretability-through-prototypes (Chen et al. (2019); Angelov & Soares (2020)); therefore it would help improve the analysis of decision making and give the problem owners agency to shape up and analyse the desired solution.

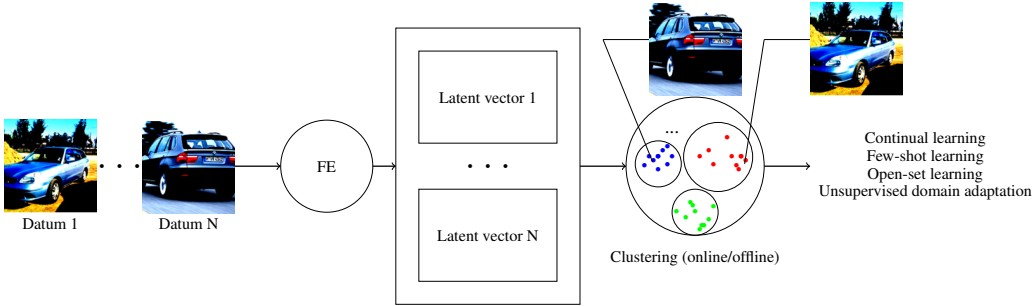

Figure 1: The overall scheme of the proposed framework (with examples from Stanford Cars dataset (Krause et al. (2013))

The proposed methodology, presented in Figure 1, involves three stages:

- extracting the latent (foundation) feature space;
- performing clustering within it;
- post-processing of the clustering results to make the overall prediction.

We show that such simple baseline, with a number of parameters multiple orders of magnitude smaller than the number of parameters of the foundation models with a clear interpretability, can solve a range of problems of unsupervised and weakly supervised learning without the need in fine-tuning of the feature space assuming the latter one has good generalisation.

## 2 RELATED WORK

**Unsupervised and semi-supervised learning**   The problems of unsupervised and semi-supervised learning take many forms. They are often solved end-to-end, by jointly performing representation learning and solving the target task. Recently, the vision transformer models (ViTs) (Dosovitskiy et al. (2021)) leveraged self-supervised representation learning (Mo et al. (2023)). In this work, we take advantage of such learnt representations and believe that further advantages in self-supervised training can capitalise upon this approach. Our focus, however, is to use the existing trained representations, both demonstrating their ability to solve multiple tasks without finetuning in self-supervised way and their ability to facilitate a range of semi-supervised learning methods such as few-shot and active learning, open set learning, continual learning, unsupervised domain adaptation between others.

Van Gansbeke et al. (2020) considers the problem of clustering of images, with and without explicit knowledge of the number of classes. Similar to this work, Van Gansbeke et al. (2020) decouple (semantically meaningful) representation learning from data clustering and decision making, however, differently from our proposed approach, both stages in Van Gansbeke et al. (2020) involve gradient descent based optimisation. Vaze et al. (2021) tackles the problem of open set recognition: recognising whether the sample is from the known classes or not. The authors suggest that it is possible to use existing classifiers, as well as that improvement of quality of prediction in the closed-set scenario is correlated with the accuracy for the open-set recognition. Galil et al. (2023) considers a similar problem of out-of-distribution data detection, and uses well-known architectures such as ViT.

**Unsupervised domain adaptation**   Peng et al. (2019) describes the problem of unsupervised domain adaptation: given the labelled model trained on one domain, is it possible to transfer, without labels, such model to a different domain representing similar data. In other words, can the model, which is capable of classifying photos of objects, be altered, without any further labels, to classify hand drawings of objects of the same classes? This work is followed up by other works solving the same task such as Dinu et al. (2023).

**Continual learning**    Buzzega et al. (2020) consider the problem of general continual learning, which is a scenario where there are no defined task boundaries, and instead there is gradual distribution shift. Rajasegaran et al. (2020) considers a problem of learning multiple tasks in a task-agnostic setting (the tasks themselves are not revealed and predicted instead). Madaan et al. (2022) is centred around the problem of unsupervised continual learning in face of data shift, as well as the related problem of few-shot learning and out-of-distribution classification with $k$ nearest neighbours.

**Visual Transformers**    Building upon the applications of the attention models to the natural language processing (Vaswani et al. (2017)), vision transformers (Dosovitskiy et al. (2021)) allowed not only to improve the performance but also provide generalisation capabilities (Zhang et al. (2022)). A substantial amount of literature is devoted to the aspects of pretraining (Singh et al. (2022)), architecture design (Liu et al. (2021)), unsupervised representation learning (Oquab et al. (2023)), and task-specific finetuning of vision transformers Dai et al. (2021). Chen et al. (2021) explored the impact of the ViT architecture on supervised learning in setting comparable to ResNets.

**Transformer Adapters**    Similar yet different idea to the proposed paper is to use transformer adapters. Such models build the task-specific architecture for problems such as multitask learning Bhattacharjee et al. (2023) and dense prediction Chen et al. (2022) on top of fixed internal representation of the existing transformer models. While the challenge of transformer adapters is to train the model on top of the transformers which would effectively solve aforementioned new tasks, this work addresses the question of what tasks can be solved through existing trained vision transformers.

**Transparency and Interpretability**    The pursuit of analysis of existing deep learning models has led to a number of methods targeting aspects of transparency such as *ante hoc*, by-design, interpretability and *post hoc* explainability. The former has been manifested by the interpretable-through-prototype methods such as ProtoPNet (Chen et al. (2019)) and xDNN (Angelov & Soares (2020)), as well as explainable architectures such as B-cos (Böhle et al. (2022)). The latter has widely used methods based on the sensitivity analysis for the input data, such as GradCAM (Selvaraju et al. (2017)) and Simonyan et al. (2014).

**Clustering**    Majority of the clustering methods and techniques were established decades ago and include such methods as $k$-means (MacQueen et al. (1967)), DBSCAN (Ester et al. (1996)) between others. Many of these methods have one or more limitations concerning streaming data clustering, scalability and problem-specific parameters such as the number of clusters or thresholds required. Bayesian non-parametric methods for clustering, Ferguson (1973), provide an established tool for streaming data clustering and can help circumvent the need to specify the number of clusters. However, a number of existing approaches does not provide solutions which are scalable enough to handle large data sample and dimensionality, Zuanetti et al. (2019); Ni et al. (2020).

To take into account the geometric nature of the problem, the methods such as kernel spectral clustering, Socher et al. (2011) have taken advantage of the kernel spaces. However, despite appealing theoretical properties and ability to detect the number of clusters, the main problem with such methods are: restrictive cluster models (i.e., assumption of cluster data distributions such a Gaussian may not work well for extremely high-dimensional data), as well as long inference time and exhaustive search for cluster candidates in an online clustering scenarios.

In this work, $k$-means is used as a baseline in offline clustering tasks, and the simple geometric clustering technique, described in Section 3.1.1, is used for on- and offline clustering.

## 3    METHODOLOGY

In the following, two distinct problem statements for clustering of latent spaces are formalised: offline and online clustering.

First, let us define the notations. Consider a Hilbert space $\mathbb{X}$ with an inner product $k(\cdot, \cdot)$. In this vector space, let us consider a finite sequence of input vectors $\mathcal{X} \subset \mathbb{X}$ with a cardinality $N_{\mathcal{X}}$. In accordance with the clustering procedure, each of the vectors from $\mathcal{X}$ is assigned a cluster from a set $\mathcal{C} = \{1 \ldots N_{\mathcal{C}}\}$ according to the assignment function $c$, where $N_{\mathcal{C}}$ may or may not be known.

**Offline clustering** The formalisation of the offline clustering of the aforementioned Hilbert space is built upon Socher et al. (2011). A (dis)similarity matrix $K \in \mathbb{R}^{N_\mathcal{X} \times N_\mathcal{X}}$ for the input vectors $\mathcal{X}$ is introduced. The problem is, based on the matrix $K$ and subject to hyperparameters $\theta$ of the algorithm, to infer the number of clusters $N_\mathcal{C}$ and the cluster assignment function $c$.

**Online clustering** In this case, a streaming data input $\mathcal{X}^n = \{\mathbf{x}_1, \mathbf{x}_2, \dots \mathbf{x}_n\}$ is considered, where the data are coming one-by-one, with the model updating the clustering for the data while the data are arriving. Only order-independent clustering is considered where for any fixed $n$ the number of clusters $N_\mathcal{C}$ and the cluster assignment function $c$ is invariant to the permutation of $\mathcal{X}^n$.

## 3.1 Methods

### 3.1.1 Online clustering

While there are many clustering methods, many of them do not satisfy the following desiderata: (1) online learning for streaming data (2) fast, non-iterative, updates for online learning (3) being non-parametric, especially with no need to specify the number of clusters. The few existing methods which satisfy such desiderata, such as ELM (Baruah & Angelov (2012)), use assumptions over the ellipsoid shape of data clusters which may not be efficient for high-dimensional spaces, and therefore we decided that simpler baseline is needed for our evaluation.

We interpret the problem of clustering as finding a solution to the following combinatorial problem: find such partition of data that a point belongs to a cluster if and only if its distance to the closest point of this cluster is less than the minimum distance to any point belonging to the other cluster. Such simple baseline clustering method, in an on- and offline variants, is described in Algorithms 1 and 2, respectively. In these algorithms, we consider `WeaklyConnectedComponents` to be a function returning weakly connected component identifiers per datum $\mathcal{C}$ as well as a number of such connected components. In the online case, `UpdateWeaklyConnectedComponents` is an online version of `WeaklyConnectedComponents` (solving *dynamic connectivity problem* Henzinger & Fredman (1998)). In the worst case scenario of the former being coincident with the latter, its complexity is $O(N)$ (which we use in our experiments). Holm et al. (2001) report, however, a more efficient, polylogarithmic, complexity for the solution of this problem. We justify use of such baseline by comparing it in the experimental section on the offline tasks, where possible, with $k$-means, and showing that this method is a simple, competitively performing nonparametric clustering method.

**Data:** $\mathcal{X} = \{\mathbf{x}_1, \mathbf{x}_2, \dots \mathbf{x}_n\}$ ; distance $d(\cdot, \cdot)$
**Result:** Clustering $\mathcal{C}$, number of clusters $N_\mathcal{C}$
$\mathcal{C} \leftarrow \emptyset$;
$\mathcal{E}^0 \leftarrow \emptyset$#edges;
**for** $i \in [1 \dots n]$ **do**
   |   $\mathcal{E}^i = \mathcal{E}^{i-1} \cup \left(\mathbf{x}_i, \mathbf{x}_{\arg\min_{j \neq i \in [1\dots(i-1)]}d(\mathbf{x}_i, \mathbf{x}_j)}\right)$;
**end**
$\mathcal{C}, N_\mathcal{C} \leftarrow$ `WeaklyConnectedComponents` $(\mathcal{G}(\mathcal{X}, \mathcal{E}^n))$;

**Algorithm 1:** Batch clustering algorithm (offline)

**Data:** $\mathcal{X} = \{\mathbf{x}_1, \mathbf{x}_2, \dots \mathbf{x}_n\}$ ; distance $d(\cdot, \cdot)$
**Result:** Clustering $\mathcal{C}$, number of clusters $N_\mathcal{C}$
$\mathcal{C} \leftarrow \emptyset$;
$\mathcal{E}^0 \leftarrow \emptyset$#edges;
**for** $i \in [1 \dots n]$ **do**
   |   $\mathcal{E}^i = \mathcal{E}^{i-1} \cup \left(\mathbf{x}_i, \mathbf{x}_{\arg\min_{j \neq i \in [1\dots(i-1)]}d(\mathbf{x}_i, \mathbf{x}_j)}\right)$;
   |   $\mathcal{C}^i, N_{\mathcal{C}^i} \leftarrow$ `UpdateWeaklyConnectedComponents`
   |   $(\mathcal{C}^{i-1}, N_{\mathcal{C}^{i-1}}, \mathcal{X}^{i-1}, \mathcal{E}^{i-1}, \mathcal{G}(\mathcal{X}^i, \mathcal{E}^i))$;
**end**

**Algorithm 2:** Clustering algorithm (online)

### 3.1.2 Clustering and Open Set recognition

We address the open set problem presented in Vaze et al. (2021): assigning the new data into the existing classes or into the none-of-the-above category. The samples which have no class label are considered outliers and assigned an 'unknown' label. Consider the clustering $\mathcal{C} = \{c_1, c_2, \ldots c_N\}$. For each cluster $c$ with the points $\mathbf{x}_i^c, i \in \{1, \ldots, n^c\}$, we use Gaussian kernel density estimation (see Figure 2) as per Parzen (1962), where $\Theta_K$ are the kernel parameters:

$$f^c(\mathbf{x}) = \frac{1}{n^c} \sum_{i=1}^{n^c} K(\mathbf{x} - \mathbf{x}_i^c | \Theta_K),$$  (1)

We compute the outliers by thresholding the maximum density value across all clusters.

### 3.1.3 Clustering and Unsupervised domain adaptation

We consider the following setting for the task: the training of the model is performed in a few-shot learning setting in the source domain, where only the labels of the prototypes close to the cluster centre are selected (see the implementation in Section 4.2); in the destination domain, only unsupervised learning is carried out (See Figure 3). We match these two clusterings of the source and target domains, $\mathcal{C}^S$ and $\mathcal{C}^T$, by the means, and assign the source domain's cluster label to the target cluster:

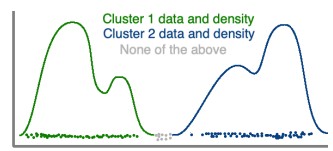

Figure 2: Clustering and Open Set Recognition

$$\forall c^2 \in C^2 \; \text{CLASS}(c^2) = \text{CLASS}(\arg\min_j l^2(\mu(c_j^1), \mu(c^2))),$$  (2)

where $\mu$ is the cluster mean operator, and CLASS is the operator of cluster's assigned class label.

## 4 Experiments

We conduct experiments on tasks such as unsupervised, continual and few-shot learning, unsupervised domain adaptation, and demonstrate that, in many cases, the methods achieve competitive generalisation without finetuning. For the experimental conditions see Appendix A.

### 4.1 Unsupervised offline learning

This setting involves clustering of the training, testing and both folds of the datasets without any knowledge of labels, except for the number of classes. The results in Table 1 and explanded results in Table 5 (Appendix) are presented for CIFAR-10, CIFAR-100, Stanford Cars and Oxford-IIIT Pets. We also present, for the reference, the baseline of the purpose-trained ResNet networks (He et al. (2016)). We evaluate the accuracy by using the Hungarian algorithm (Egerváry (1931)), following the procedure described in Appendix A.1. For this simple experiment, we assume the known

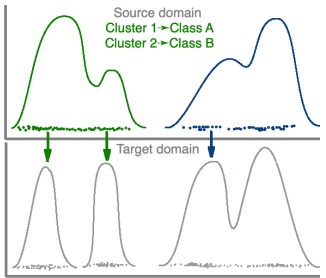

Figure 3: Unsupervised domain adaptation

number of clusters, coincident with the number of classes, and report the results on the training, testing and both folds. We see that the results between different folds do not differ much, while the choice of architecture (using of larger architectures and/or finetuning on ImageNet-1K) makes a substantial difference, with finetuned on ImageNet-1K (and not on a target dataset) results giving, notably, over $97\%$ on CIFAR-10 without any label information and $91\%$ on Oxford-IIIT Pet.

---

[1] From https://github.com/weiaicunzai/pytorch-cifar100

[2] From https://github.com/matkalinowski/EfficientNet-on-Stanford-Cars-Dataset

[3] From https://github.com/matkalinowski/EfficientNet-on-Stanford-Cars-Dataset

| FE | method | accuracy, training/testing/both (%) |
|---|---|---|
| CIFAR-10 | | |
| ViT-H/14 (SWAG) | $k$-means | $81.66 \pm 7.40 / 85.18 \pm 7.42 / 88.89 \pm 6.28$ |
| ViT-H/14 (SWAG+IN1K) | $k$-means | $97.05 \pm 0.00 / 97.03 \pm 0.03 / 97.06 \pm 0.00$ |
| SCAN Van Gansbeke et al. (2020) | SCAN | $87.6 \pm 0.4$ (testing) |
| ResNet-110 (He et al. (2016)) | - | $93.39 \pm 0.16$ (testing) |
| CIFAR-100 | | |
| ViT-H/14 (SWAG) | $k$-means | $55.85 \pm 0.74 / 54.40 \pm 1.48 / 55.08 \pm 0.77$ |
| ViT-H/14 (SWAG+IN1K) | $k$-means | $71.94 \pm 0.86 / 72.10 \pm 0.96 / 72.80 \pm 0.42$ |
| RESNET-152(He et al. (2016))[1] | - | 77.69 (testing) |
| Stanford Cars | | |
| ViT-H/14 (SWAG) | $k$-means | $44.76 \pm 0.57 / 44.12 \pm 0.52 / 44.64 \pm 0.66$ |
| ViT-H/14 (SWAG+IN1K) | $k$-means | $48.55 \pm 0.76 / 48.42 \pm 0.85 / 49.24 \pm 0.63$ |
| RESNET-152(He et al. (2016))[2] | - | 77.8 (testing) |
| Oxford-IIIT Pet | | |
| ViT-H/14 (SWAG) | $k$-means | $87.55 \pm 2.79 / 86.10 \pm 2.56 / 87.62 \pm 2.47$ |
| ViT-H/14 (SWAG+IN1K) | $k$-means | $90.35 \pm 2.05 / 91.59 \pm 1.21 / 91.01 \pm 2.45$ |
| RESNET-152(He et al. (2016))[3] | - | 94.5 (testing) |

Table 1: Clustering results (CIFAR10&100, Stanford Cars/Oxford-IIIT Pet); SWAG denotes weakly supervised SWAG (Singh et al. (2022)), IN1K denotes finetuning on ImageNet-1K (5 runs), number of clusters exactly matches the number of classes, ResNet models are a supervised learning baseline

| FE | method | clusters | accuracy (training/testing) (%) |
|---|---|---|---|
| CIFAR-10 | | | |
| ViT-H/14 (SWAG) | $k$-means | 200 | $95.31 \pm 0.20 / 95.15 \pm 0.16$ |
| ViT-H/14 (SWAG) | $k$-means | 10 | $84.26 \pm 6.06 / 81.41 \pm 4.85$ |
| ViT-H/14 (SWAG) | batch | 4092 | $96.66 / 96.40$ |
| ViT-H/14 (SWAG+IN1K) | $k$-means | 200 | $96.42 \pm 0.17 / 96.54 \pm 0.13$ |
| ViT-H/14 (SWAG+IN1K) | $k$-means | 10 | $97.05 \pm 0.00 / 93.62 \pm 0.00$ |
| ViT-H/14 (SWAG+IN1K) | batch | 5896 | $97.67 / 97.05$ |
| CIFAR-100 | | | |
| ViT-H/14 (SWAG) | $k$-means | 2000 | $72.66 \pm 0.36 / 71.01 \pm 0.50$ |
| ViT-H/14 (SWAG) | $k$-means | 100 | $55.35 \pm 0.62 / 54.77 \pm 0.64$ |
| ViT-H/14 (SWAG) | batch | 4029 | $72.95 / 73.31$ |
| ViT-H/14 (SWAG+IN1K) | $k$-means | 2000 | $80.72 \pm 0.11 / 80.46 \pm 0.12$ |
| ViT-H/14 (SWAG+IN1K) | $k$-means | 100 | $72.52 \pm 0.84 / 72.97 \pm 0.60$ |
| ViT-H/14 (SWAG+IN1K) | batch | 6313 | $84.28 / 81.84$ |

Table 2: Few shot active learning results (CIFAR-10 and CIFAR100); SWAG denotes weakly supervised SWAG (Singh et al. (2022)) pretraining, IN1K denotes finetuning on ImageNet-1K

## 4.2 FEW-SHOT ACTIVE LEARNING

For the few-shot learning task, we select, for each cluster centroid, the closest real example. We assume that for these examples we can request labels. The results in Table 2, expanded in Tables 6 and 7, show substantial improvement in performance comparing to the unsupervised learning in the previous section: while the best results for CIFAR-100 reach $84\%$ in this setting, one can get only $72\%$ in the clustering scenario. When taking into account the number of clusters, however, the equivalent one-shot active learning scenario yields comparable $72.97\%$ accuracy against $72.80\%$ in the unsupervised case. Hereafter we refer to the implementation of Algorithm 1 as 'batch'.

| no f/t, 1 task at a time | no f/t, 10 tasks at a time | f/t on IN1K, 1 task at a time | f/t on IN1K, 10 tasks at a time |

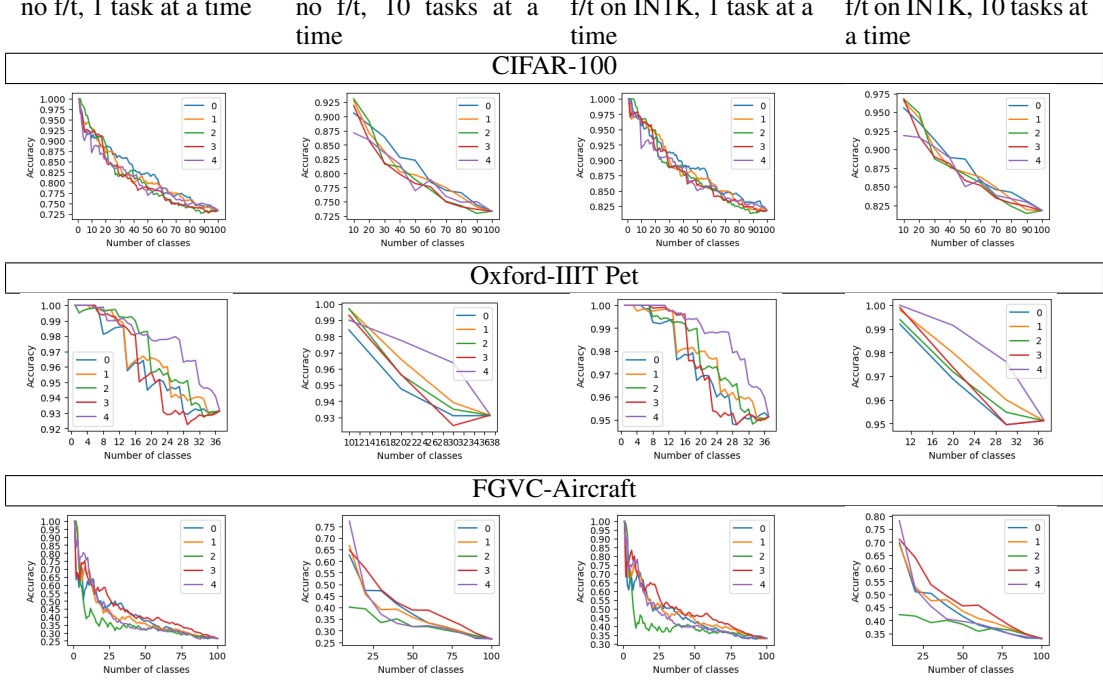

Figure 4: Online learning, ViT-H/14, f/t=finetuning, run on five randomly shuffled sequences

## 4.3 WEAKLY-SUPERVISED TASK AGNOSTIC CONTINUAL ACTIVE LEARNING

Following the experimental setting from Rajasegaran et al. (2020) and using Algorithm 2 for online clustering, we provide the continual learning results in Figure 4. The end performance, by design of the online algorithm, exactly coincides with the corresponding batch results as per Section 4.2. We present additional testing scenarios in Appendix E. For CIFAR-100, the model with a task increment of ten achieves 81.84%, while the incremental learning iTAML model, using a reduced version of ResNet-18 (ResNet-18(1/3)) and gradient descent based incremental learning method, achieves 78% according to Figure 6 of Rajasegaran et al. (2020). The naïve finetuning in the same scenario gives circa 12% (Rajasegaran et al. (2020)). This results support the hypothesis that it is possible to match the performance of the best purpose-built existing continual learning methods by using the fixed feature space without finetuning, if such feature space (such as ViT-H/14) is rich enough.

## 4.4 OPEN-SET RECOGNITION

The results in Table 3 compare the open-set recognition method, described in Section 3.1.2, to the baselines according to Vaze et al. (2021). We notice that, consistently with the few-shot and online learning results, described in Sections 4.1 and 4.2, the results on both closed and open set are worse for the fine-grained classification problems, which is especially noticeable for the FGVC-Aircraft data. As detailed in Appendix A.2, we employ the same split as per Vaze et al. (2021): the accuracy is reported for the closed set, AUROC and OSCR are reported on easy / medium&hard open classes.

## 4.5 UNSUPERVISED DOMAIN ADAPTATION

We reproduce the experiment setting from Peng et al. (2019), with the precise experimental details described in Appendix A. Our scenario differs from Peng et al. (2019) in that the model is only fitted on the source domain in a few-shot active learning setting as per Section 4.2, where only selected examples of the data from multiple domains are labelled, in order to analyse the model's ability to work in a semi-supervised setting. Domain adaptation is implemented according to the method described in Section 3.1.3. The results for the experimental setting, which uses the DomainNet dataset and batch clustering, are presented in Table 4; further examples, using ViT-H/14 without

| Task | Method | closed set acc. (%) | AUROC | OSCR |
|---|---|---|---|---|
| CUB
Welinder et al. (2010) | ARPL+(from Vaze et al. (2021)) | 85.9 | 83.5/75.5 | 76.0/69.6 |
| | MLS(from Vaze et al. (2021)) | 88.3 | 88.3/79.3 | 79.8/73.1 |
| | proposed | 92.35 | 88.95/75.84 | 79.22/68.72 |
| Stanford Cars
Krause et al. (2013) | ARPL+(from Vaze et al. (2021)) | 96.9 | 94.8/83.6 | 92.8/82.3 |
| | MLS(from Vaze et al. (2021)) | 97.1 | 94.0/82.2 | 92.2/81.1 |
| | proposed | 85.23 | 76.87/71.03 | 62.06/57.62 |
| FGVC-Aircraft
Maji et al. (2013) | ARPL+(from Vaze et al. (2021)) | 91.5 | 87.0/77.7 | 83.3/74.9 |
| | MLS(from Vaze et al. (2021)) | 91.7 | 90.7/82.3 | 86.8/79.8 |
| | proposed | 37.54 | 63.97/51.08 | 24.12/18.77 |

Table 3: Open set task performance (batch clustering, VIT-H/14, SWAG pretrained on ImageNet1K), the # of clusters is 437, 707 and 506 for CUB, Stanford Cars, and FGVC-Aircraft respectively

| | clp | inf | pnt | qdr | rel | skt | avg | best baseline avg | oracle |
|---|---|---|---|---|---|---|---|---|---|
| clp | 79.72 | 62.71 | 66.01 | 44.88 | 71.75 | 70.73 | 63.21 | 24.1 | $71.0 \pm 0.63$ |
| inf | 37.24 | 50.12 | 32.73 | 25.51 | 38.55 | 36.08 | 34.02 | 20.2 | $36.1 \pm 0.61$ |
| pnt | 62.75 | 56.00 | 74.03 | 37.48 | 65.75 | 62.06 | 56.81 | 26.0 | $68.1 \pm 0.49$ |
| qdr | 12.24 | 9.37 | 8.61 | 35.19 | 9.91 | 12.61 | 10.55 | 9.3 | $69.1 \pm 0.52$ |
| rel | 77.49 | 71.51 | 74.08 | 43.04 | 87.29 | 75.73 | 68.37 | 27.2 | $81.3 \pm 0.49$ |
| skt | 61.00 | 53.30 | 58.00 | 37.26 | 59.72 | 70.00 | 53.86 | 24.2 | $65.2 \pm 0.57$ |
| avg | 50.14 | 50.58 | 47.89 | 37.63 | 49.14 | 51.44 | 47.80 | 21.9 | $65.1 \pm 0.55$ |

Table 4: Unsupervised domain adaptation, ViT-H/14, finetuned on ImageNet1K (columns denote source domains, and rows denote target domains), the oracle values are taken from Peng et al. (2019) for ResNet-152, best baseline is MCD (Saito et al. (2018)), taken from Peng et al. (2019)

finetuning and k-means clustering, are presented in Appendix D. It contains six domains (namely sketch (ske), real (rel), quickdraw (qdr), painting (pnt), infograph (inf), and clipart(clp)) , each split into the same 345 categories of common objects. As we demonstrate further in Section 4.6, not only the method performs much better in scenarios of single domain adaptation from Peng et al. (2019), it also can demonstrate, through the interpretation mechanism, justification for decision making. The oracle baseline corresponds to the setting of supervised training on the target domain.

## 4.6 ANALYSIS OF INTERPRETABILITY

We demonstrate that not only the method constitutes a strong baseline, but it is also capable of interpretability through prototypes. Apart from the visual power and appeal to human perception of the prototypes the method also provides the distance between a query and the nearest prototype (see Figure 5) which is a numerical estimate of the dissimilarity. For the same latent space, this numerical estimate can be compared like-for-like between different examples and datasets.

One can see from Figure 5, as well as from Figure 8 in the Appendix, that in many cases, even for the incorrect classification, the closest examples make semantic sense. In Figure 8a, the model successfully relates aircraft carriers in different poses, sketched or photographed. The alarm clock (Figure 8b) recognises the quickdraw prototypes, but struggles to generalise to the sketch prototypes. Angel quickdraw prototypes (see Figure 5a ) show confusion (in the quickdraw domain) between similarly-looking angels, spiders and bats. The results are worse for cross-domain examples. The quickdraw sample in Figure 5b, supposed to be a bat, looks indeed much like an apple. This is duly reflected by the model outputs. Coffee cup example (see Figure 5c), demonstrates competition between the coffee cup and the knife prototypes as the image contains both. Seeing the nearest blueberry example to the real diamond query image (see Figure 5d) helps make sense out of the incorrect recognition. In the hot tub example (Figure 8c), one can see that the varieties of appearances of the same objects are shown to have a low distance. The feature space also places closer different types of light fixtures, as one can see in Figure 8d, in sketch and real domains alike.The limitations of differentiating between geometric figures such as hexagons and octagons (see Figure 5e suggest

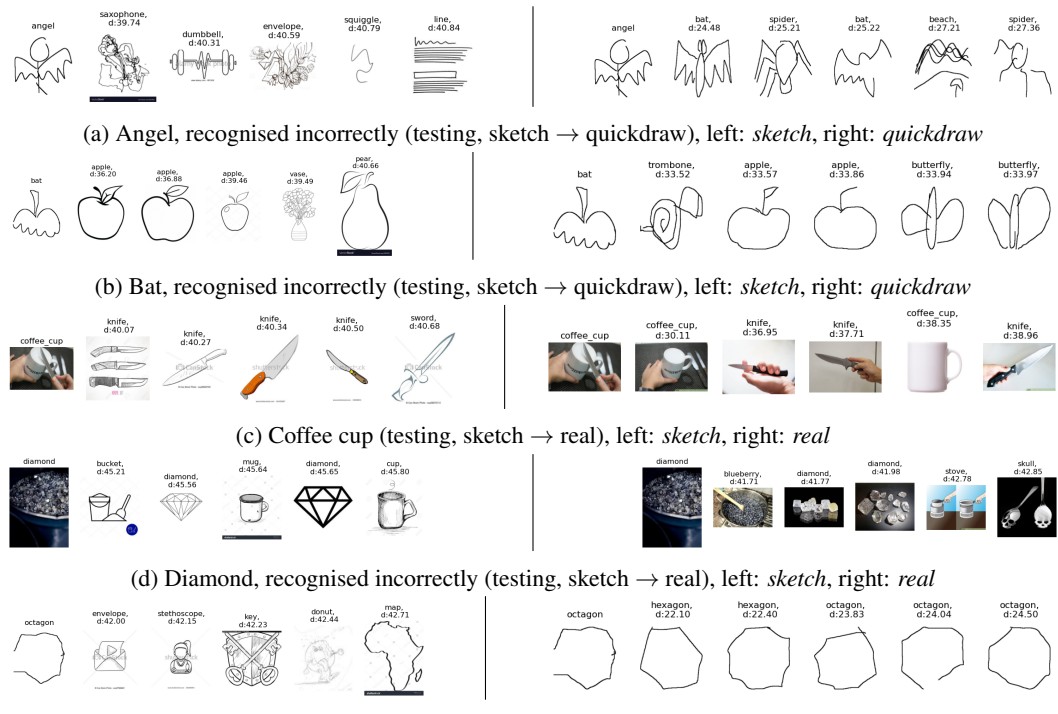

(a) Angel, recognised incorrectly (testing, sketch → quickdraw), left: *sketch*, right: *quickdraw*

(b) Bat, recognised incorrectly (testing, sketch → quickdraw), left: *sketch*, right: *quickdraw*

(c) Coffee cup (testing, sketch → real), left: *sketch*, right: *real*

(d) Diamond, recognised incorrectly (testing, sketch → real), left: *sketch*, right: *real*

(e) Octagon, recognised incorrectly (testing, sketch → quickdraw), left: *sketch*, right: *quickdraw*

Figure 5: Interpretability examples (the leftmost images are queries, and the further ones are prototypes)

that despite generality, textural information often trumps the semantic one. For the dogs scenario on training data (see Figure 8e, one can see that while the model has sometimes advantage to find the exact match with the prototypes, it also shows remarkable cross-domain generalisation.

Figure 9 of the Appendix demonstrates that the inferior results for datasets such as Stanford Cars (Krause et al. (2013)) is caused by difficulty to distinguish between similar fine-grained classes. While the foundational latent spaces allow remarkable generalisation, they still have limitations of accuracy on such problems. Therefore, further representation learning refinement is necessary.

## 5 CONCLUSION

We show that, with just fixed foundational feature representation and an online clustering technique, one can solve a number of problems such as open set recognition, unsupervised and few-shot learning, continual learning and unsupervised domain adaptation, in a way that is interpretable through prototypes. We conclude that while the results are strong on coarse-grained datasets such as CIFAR-100, they are lagging behind in fine-grained problems such as Stanford Cars and FGVC-Aircraft (see Table 3). In many cases, however, even though the results are wrong from the benchmark's point of view, they still can constitute a plausible answer (see Figures 5 and 9). These problems have potential to be solved by improvements in fixed representation learning through foundational feature extractors without finetuning, which becomes, as it is evidenced by the experiments above, a plausible alternative to finetuning-based continual learning methods. The described methodology can serve as a benchmark for evaluating such representation learning.

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

## A EXPERIMENTAL SETUP

We use the pretrained SWAG-ViT (Singh et al. (2022)) models from publicly available repository[4]. H/14 (SWAG) model corresponds to the pretrained model with no finetuning. H/14(SWAG+IN1K) model corresponds to the model finetuned on ImageNet1K. The same convention applies to different variations of SWAG-ViT: ViT-L/16, as well as ViT-B/16. To ensure consistent experimental environment, for all experiments we use NVIDIA RTX A2000 12GB powered workstation.

sklearn $k$-means clustering is used. The batch clustering is implemented using sklearn and pytorch. For the kernel density estimator in Section 4.4, we use `sklearn.neighbors.KernelDensity` class with a bandwidth of 0.2.

For the unsupervised (Section 4.1) and few-shot learning (Section 4.2) experiments, we estimate confidence intervals by running the clustering methods five different times with five different randoms seeds. The proposed batch clustering baseline is deterministic, and therefore the confidence intervals could not be given.

### A.1 HUNGARIAN ALGORITHM EVALUATION OF UNSUPERVISED LEARNING

The evaluation of the unsupervised learning in Section 4.1 is performed using the Hungarian algorithm matching: the cluster identifiers are matched to the ground-truth labels by maximising the sum of number of samples in the clusters corresponding to the class given unique one-to-one assignment between the clusters and the classes.

### A.2 OPEN SET TRAINING CONDITIONS

We replicate the same open-set evaluation protocol as Vaze et al. (2021). The split of data is the same. The accuracy is reported for the closed set, AUROC and Open-Set Classification Rate (OSCR) are reported on easy / medium&hard open classes. OSCR Dhamija et al. (2018) is designed to measure the trade-off between accuracy and detection rate for varying confidence threshold.

---

[4]https://github.com/facebookresearch/SWAG

| FE | method | accuracy, training/testing/both (%) |
|---|---|---|
| CIFAR-10 | | |
| VIT-B/16 (SWAG) | $k$-means | $73.72 \pm 6.97 / 76.29 \pm 3.92 / 74.30 \pm 5.00$ |
| VIT-B/16 (SWAG+IN1K) | $k$-means | $91.15 \pm 1.68 / 91.56 \pm 1.42 / 91.21 \pm 1.65$ |
| VIT-L/16 (SWAG) | $k$-means | $81.48 \pm 2.72 / 83.05 \pm 4.92 / 86.04 \pm 6.00$ |
| VIT-L/16 (SWAG+IN1K) | $k$-means | $89.56 \pm 1.87 / 87.42 \pm 2.76 / 89.57 \pm 1.90$ |
| VIT-H/14 (SWAG) | $k$-means | $81.66 \pm 7.40 / 85.18 \pm 7.42 / 88.89 \pm 6.28$ |
| VIT-H/14 (SWAG+IN1K) | $k$-means | $97.05 \pm 0.00 / 97.03 \pm 0.03 / 97.06 \pm 0.00$ |
| SCAN Van Gansbeke et al. (2020) | SCAN | $87.6 \pm 0.4$ (testing) |
| ResNet-110 (He et al. (2016)) | - | $93.39 \pm 0.16$ (testing) |
| CIFAR-100 | | |
| VIT-B/16 (SWAG) | $k$-means | $44.09 \pm 0.49 / 43.48 \pm 0.70 / 43.64 \pm 0.28$ |
| VIT-L/16 (SWAG) | $k$-means | $55.36 \pm 1.25 / 53.20 \pm 1.15 / 54.53 \pm 1.17$ |
| VIT-L/16 (SWAG+IN1K) | $k$-means | $68.06 \pm 1.42 / 67.88 \pm 0.91 / 68.09 \pm 0.87$ |
| VIT-H/14 (SWAG) | $k$-means | $55.85 \pm 0.74 / 54.40 \pm 1.48 / 55.08 \pm 0.77$ |
| VIT-H/14 (SWAG+IN1K) | $k$-means | $71.94 \pm 0.86 / 72.10 \pm 0.96 / 72.80 \pm 0.42$ |
| RESNET-152[5] | - | 77.69 (testing) |
| Stanford Cars | | |
| VIT-H/14 (SWAG) | $k$-means | $44.76 \pm 0.57 / 44.12 \pm 0.52 / 44.64 \pm 0.66$ |
| VIT-H/14 (SWAG+IN1K) | $k$-means | $48.55 \pm 0.76 / 48.42 \pm 0.85 / 49.24 \pm 0.63$ |
| RESNET-152[6] | - | 77.8 |
| Oxford-IIIT Pet | | |
| VIT-H/14 (SWAG) | $k$-means | $87.55 \pm 2.79 / 86.10 \pm 2.56 / 87.62 \pm 2.47$ |
| VIT-H/14 (SWAG+IN1K) | $k$-means | $90.35 \pm 2.05 / 91.59 \pm 1.21 / 91.01 \pm 2.45$ |
| RESNET-152[7] | - | 94.5 (testing) |

Table 5: Clustering results (CIFAR10&100, Stanford Cars/Oxford-IIIT Pet); SWAG denotes weakly supervised SWAG (Singh et al. (2022)), IN1K denotes finetuning on ImageNet-1K (5 runs), number of clusters exactly matches the number of classes

## B    EXPANDED CLUSTERING RESULTS

We present the expanded clustering results in Table 5.

## C    EXPANDED RESULTS ON FEW SHOT ACTIVE LEARNING

We present additional results for few shot active learning in Table 6

## D    EXPANDED RESULTS ON UNSUPERVISED DOMAIN ADAPTATION

In Tables 8 and 9, we present further results on the problem of unsupervised domain adaptation.

## E    FURTHER RESULTS ON ONLINE LEARNING

In Figures 6 and 7, we present further results on online learning.

## F    FURTHER INTERPRETABILITY EXAMPLES

Further interpretability examples, discussed in Section 4.6, presented in Figures 8 and 9.

| FE | method | clusters | accuracy (training/testing) (%) |
|---|---|---|---|
| ViT-B/16 (SWAG) | $k$-means | 200 | $87.98 \pm 0.49 / 86.78 \pm 0.38$ |
| ViT-B/16 (SWAG) | $k$-means | 100 | $86.86 \pm 0.80 / 85.30 \pm 0.92$ |
| ViT-B/16 (SWAG) | $k$-means | 50 | $85.25 \pm 0.58 / 83.36 \pm 0.29$ |
| ViT-B/16 (SWAG) | $k$-means | 10 | $73.06 \pm 5.02 / 71.17 \pm 4.42$ |
| ViT-B/16 (SWAG) | batch | 4700 | $89.74 / 90.71$ |
| ViT-B/16 (SWAG+IN1K) | $k$-means | 200 | $92.54 \pm 0.47 / 92.00 \pm 0.29$ |
| ViT-B/16 (SWAG+IN1K) | $k$-means | 100 | $92.00 \pm 0.49 / 91.32 \pm 0.31$ |
| ViT-B/16 (SWAG+IN1K) | $k$-means | 50 | $91.78 \pm 0.76 / 91.13 \pm 0.65$ |
| ViT-B/16 (SWAG+IN1K) | $k$-means | 10 | $91.15 \pm 0.17 / 88.59 \pm 0.25$ |
| ViT-B/16 (SWAG+IN1K) | batch | 6363 | $95.37 / 94.10$ |
| ViT-L/16 (SWAG) | $k$-means | 200 | $94.76 \pm 0.21 / 94.57 \pm 0.06$ |
| ViT-L/16 (SWAG) | $k$-means | 100 | $93.99 \pm 0.11 / 93.64 \pm 0.19$ |
| ViT-L/16 (SWAG) | $k$-means | 50 | $92.71 \pm 0.66 / 92.66 \pm 0.56$ |
| ViT-L/16 (SWAG) | $k$-means | 10 | $84.68 \pm 1.18 / 82.68 \pm 0.63$ |
| ViT-L/16 (SWAG) | batch | 4444 | $95.83 / 95.75$ |
| ViT-L/16 (SWAG+IN1K) | $k$-means | 200 | $94.95 \pm 0.15 / 94.75 \pm 0.09$ |
| ViT-L/16 (SWAG+IN1K) | $k$-means | 100 | $94.34 \pm 0.51 / 94.69 \pm 0.27$ |
| ViT-L/16 (SWAG+IN1K) | $k$-means | 50 | $93.58 \pm 0.29 / 94.19 \pm 0.18$ |
| ViT-L/16 (SWAG+IN1K) | $k$-means | 10 | $89.56 \pm 1.87 / 91.50 \pm 0.16$ |
| ViT-L/16 (SWAG+IN1K) | batch | 5983 | $96.79 / 95.69$ |
| ViT-H/14 (SWAG) | $k$-means | 200 | $95.31 \pm 0.20 / 95.15 \pm 0.16$ |
| ViT-H/14 (SWAG) | $k$-means | 100 | $95.27 \pm 0.20 / 94.63 \pm 0.32$ |
| ViT-H/14 (SWAG) | $k$-means | 50 | $94.38 \pm 0.38 / 93.25 \pm 0.20$ |
| ViT-H/14 (SWAG) | $k$-means | 10 | $84.26 \pm 6.06 / 81.41 \pm 4.85$ |
| ViT-H/14 (SWAG) | batch | 4092 | $96.66 / 96.40$ |
| ViT-H/14 (SWAG+IN1K) | $k$-means | 200 | $96.42 \pm 0.17 / 96.54 \pm 0.13$ |
| ViT-H/14 (SWAG+IN1K) | $k$-means | 100 | $95.93 \pm 0.11 / 96.35 \pm 0.21$ |
| ViT-H/14 (SWAG+IN1K) | $k$-means | 50 | $96.09 \pm 0.25 / 96.26 \pm 0.21$ |
| ViT-H/14 (SWAG+IN1K) | $k$-means | 10 | $97.05 \pm 0.00 / 93.62 \pm 0.00$ |
| ViT-H/14 (SWAG+IN1K) | batch | 5896 | $97.67 / 97.05$ |

Table 6: Few shot active learning results (CIFAR10); SWAG denotes weakly supervised SWAG (Singh et al. (2022)) pretraining, IN1K denotes finetuning on ImageNet-1K

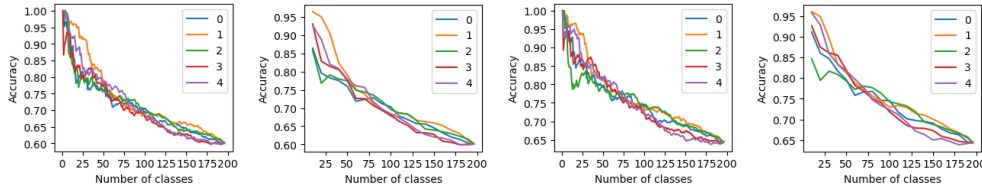

(a) ViT-H/14 (no f/t), 1 task at a time  (b) ViT-H/14 (no f/t), 10 tasks at a time  (c) ViT-H/14 (f/t on IN1K), 1 task at a time  (d) ViT-H/14 (f/t on IN1K), 10 tasks at a time

Figure 6: Online learning for Stanford Cars task

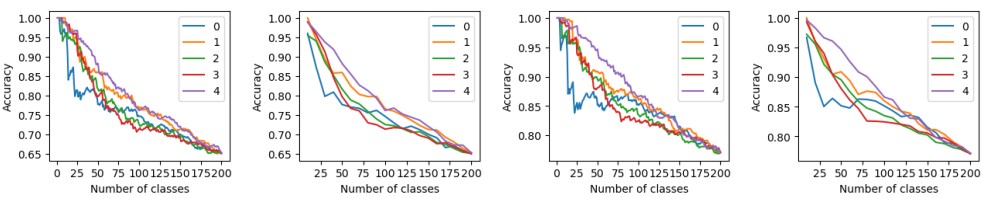

(a) ViT-H/14 (no f/t), 1 task at a time  (b) ViT-H/14 (no f/t), 10 tasks at a time  (c) ViT-H/14 (f/t on IN1K), 1 task at a time  (d) ViT-H/14 (f/t on IN1K), 10 tasks at a time

Figure 7: Online learning for CUB task

| FE | method | clusters | accuracy (%) |
|---|---|---|---|
| VɪT-B/16 (SWAG) | $k$-means | 2000 | $59.53 \pm 0.14 / 56.88 \pm 0.16$ |
| VɪT-B/16 (SWAG) | $k$-means | 1000 | $57.02 \pm 0.26 / 54.81 \pm 0.78$ |
| VɪT-B/16 (SWAG) | $k$-means | 500 | $53.92 \pm 0.71 / 51.84 \pm 0.27$ |
| VɪT-B/16 (SWAG) | $k$-means | 100 | $42.17 \pm 0.64 / 41.56 \pm 0.32$ |
| VɪT-B/16 (SWAG) | batch | 4625 | $59.09 / 60.13$ |
| VɪT-B/16 (SWAG+IN1K) | $k$-means | 2000 | $72.12 \pm 0.28 / 69.84 \pm 0.42$ |
| VɪT-B/16 (SWAG+IN1K) | $k$-means | 1000 | $70.36 \pm 0.24 / 68.34 \pm 0.21$ |
| VɪT-B/16 (SWAG+IN1K) | $k$-means | 500 | $67.96 \pm 0.50 / 66.05 \pm 0.58$ |
| VɪT-B/16 (SWAG+IN1K) | $k$-means | 100 | $60.34 \pm 0.85 / 57.55 \pm 0.52$ |
| VɪT-B/16 (SWAG+IN1K) | batch | 6719 | $76.98 / 73.58$ |
| VɪT-L/16 (SWAG) | $k$-means | 2000 | $70.74 \pm 0.30 / 69.02 \pm 0.19$ |
| VɪT-L/16 (SWAG) | $k$-means | 1000 | $67.83 \pm 0.33 / 66.98 \pm 0.41$ |
| VɪT-L/16 (SWAG) | $k$-means | 500 | $65.43 \pm 0.34 / 65.21 \pm 0.28$ |
| VɪT-L/16 (SWAG) | $k$-means | 100 | $54.90 \pm 0.75 / 54.02 \pm 0.84$ |
| VɪT-L/16 (SWAG) | batch | 4319 | $72.13 / 70.91$ |
| VɪT-L/16 (SWAG+IN1K) | $k$-means | 2000 | $77.79 \pm 0.10 / 76.95 \pm 0.13$ |
| VɪT-L/16 (SWAG+IN1K) | $k$-means | 1000 | $75.82 \pm 0.13 / 75.78 \pm 0.29$ |
| VɪT-L/16 (SWAG+IN1K) | $k$-means | 500 | $74.04 \pm 0.27 / 74.43 \pm 0.26$ |
| VɪT-L/16 (SWAG+IN1K) | $k$-means | 100 | $67.80 \pm 1.25 / 68.76 \pm 0.79$ |
| VɪT-L/16 (SWAG+IN1K) | batch | 6317 | $81.79 / 78.89$ |
| VɪT-H/14 (SWAG) | $k$-means | 2000 | $72.66 \pm 0.36 / 71.01 \pm 0.50$ |
| VɪT-H/14 (SWAG) | $k$-means | 1000 | $69.92 \pm 0.36 / 69.38 \pm 0.33$ |
| VɪT-H/14 (SWAG) | $k$-means | 500 | $67.59 \pm 0.43 / 67.20 \pm 0.50$ |
| VɪT-H/14 (SWAG) | $k$-means | 100 | $55.35 \pm 0.62 / 54.77 \pm 0.64$ |
| VɪT-H/14 (SWAG) | batch | 4029 | $72.95 / 73.31$ |
| VɪT-H/14 (SWAG+IN1K) | $k$-means | 10000 | $89 / 82$ |
| VɪT-H/14 (SWAG+IN1K) | $k$-means | 5000 | $84 / 82$ |
| VɪT-H/14 (SWAG+IN1K) | $k$-means | 2000 | $80.72 \pm 0.11 / 80.46 \pm 0.12$ |
| VɪT-H/14 (SWAG+IN1K) | $k$-means | 1000 | $79.13 \pm 0.40 / 79.51 \pm 0.56$ |
| VɪT-H/14 (SWAG+IN1K) | $k$-means | 500 | $77.87 \pm 0.19 / 78.75 \pm 0.25$ |
| VɪT-H/14 (SWAG+IN1K) | $k$-means | 100 | $72.52 \pm 0.84 / 72.97 \pm 0.60$ |
| VɪT-H/14 (SWAG+IN1K) | batch | 6313 | $84.28 / 81.84$ |

Table 7: Few shot active learning results(CIFAR100); SWAG denotes weakly supervised SWAG (Singh et al. (2022)) pretraining, IN1K denotes finetuning on ImageNet-1K

| | clp | inf | pnt | qdr | rel | skt | avg |
|---|---|---|---|---|---|---|---|
| clp | 0.7446 | 0.4167 | 0.4907 | 0.4247 | 0.5991 | 0.6001 | 0.5063 |
| inf | 0.3358 | 0.4267 | 0.2931 | 0.2197 | 0.3537 | 0.3242 | 0.3053 |
| pnt | 0.5068 | 0.3690 | 0.6777 | 0.3596 | 0.5230 | 0.5197 | 0.4556 |
| qdr | 0.0960 | 0.0452 | 0.0562 | 0.3639 | 0.0798 | 0.0809 | 0.0716 |
| rel | 0.6903 | 0.5734 | 0.6111 | 0.4256 | 0.8290 | 0.6622 | 0.5925 |
| skt | 0.5701 | 0.3886 | 0.4887 | 0.4051 | 0.5269 | 0.6820 | 0.4759 |
| avg | 0.4398 | 0.3586 | 0.3880 | 0.3669 | 0.4165 | 0.4374 | 0.4012 |

Table 8: Unsupervised domain adaptation, ViT-H/14 without finetuning (columns denote source domains, and rows denote target domains)

|  | clp | inf | pnt | qdr | rel | skt | avg |
|---|---|---|---|---|---|---|---|
| clp | $77.92 \pm 0.08$ | $58.05 \pm 1.57$ | $64.02 \pm 0.68$ | $39.21 \pm 0.62$ | $68.97 \pm 0.16$ | $68.39 \pm 0.37$ | $59.73 \pm 0.28$ |
| inf | $37.20 \pm 0.62$ | $47.39 \pm 0.44$ | $29.94 \pm 1.61$ | $22.97 \pm 0.46$ | $34.21 \pm 0.36$ | $34.20 \pm 0.94$ | $31.70 \pm 0.38$ |
| pnt | $63.46 \pm 0.46$ | $54.06 \pm 1.70$ | $71.77 \pm 0.27$ | $34.91 \pm 0.88$ | $64.31 \pm 0.29$ | $61.55 \pm 0.27$ | $55.66 \pm 0.36$ |
| qdr | $9.92 \pm 0.06$ | $7.54 \pm 0.63$ | $6.34 \pm 0.19$ | $28.21 \pm 0.18$ | $7.30 \pm 0.26$ | $10.54 \pm 0.18$ | $8.33 \pm 0.18$ |
| rel | $77.05 \pm 0.49$ | $66.88 \pm 1.01$ | $72.68 \pm 0.26$ | $37.29 \pm 0.83$ | $84.83 \pm 0.22$ | $73.96 \pm 0.64$ | $65.57 \pm 0.29$ |
| skt | $59.37 \pm 0.64$ | $49.68 \pm 1.55$ | $56.15 \pm 0.19$ | $33.25 \pm 0.21$ | $57.02 \pm 0.52$ | $66.43 \pm 0.12$ | $51.10 \pm 0.32$ |
| avg | $49.40 \pm 0.44$ | $47.25 \pm 1.21$ | $45.83 \pm 0.50$ | $33.53 \pm 0.53$ | $46.37 \pm 0.19$ | $49.72 \pm 0.39$ | $45.35 \pm 0.15$ |

Table 9: Unsupervised domain adaptation, ViT-H/14, $k$-means clustering ($5 \times 345 = 1725$ clusters, 345 classes), ImageNet1K finetuning (columns denote source domains, and rows denote target domains)

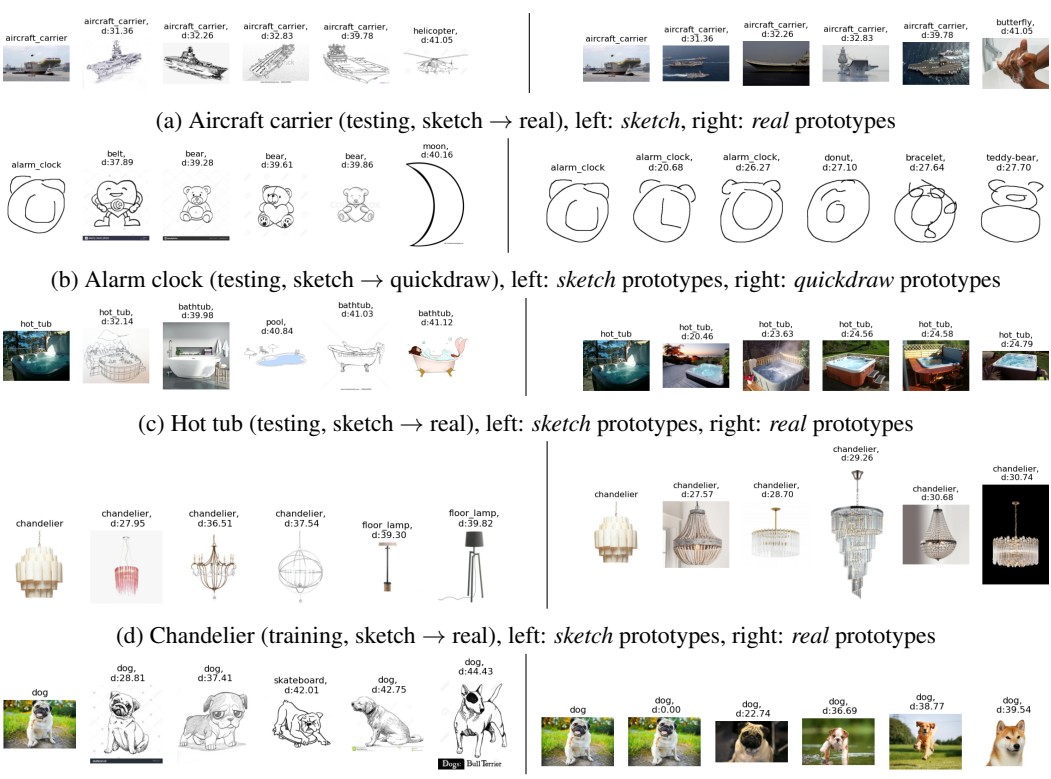

(a) Aircraft carrier (testing, sketch → real), left: *sketch*, right: *real* prototypes

(b) Alarm clock (testing, sketch → quickdraw), left: *sketch* prototypes, right: *quickdraw* prototypes

(c) Hot tub (testing, sketch → real), left: *sketch* prototypes, right: *real* prototypes

(d) Chandelier (training, sketch → real), left: *sketch* prototypes, right: *real* prototypes

(e) Dog (training, sketch → real), left: *sketch* prototypes, right: *real* prototypes

Figure 8: Interpretability examples

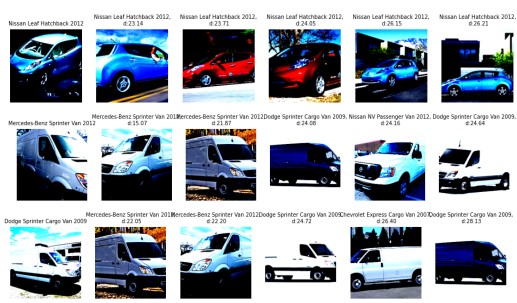

Figure 9: Interpretability examples: few shot learning (Stanford Cars Krause et al. (2013))

