# OpenReview forum: "How far can we go without finetuning?"
_ICLR.cc/2024/Conference — ICLR 2024 Conference Withdrawn Submission_

### Official Review · Reviewer_YapA · 2023-10-30

**Soundness:** 2 fair
**Presentation:** 1 poor
**Contribution:** 3 good
**Rating:** 5
**Confidence:** 4

**Summary:**

This paper investigates to what extent clustering of latent representations in pre-trained classifiers can be used to address problems such as few-shot learning, unsupervised domain adaptation and continual learning. The motivation is to provide a simple, competitive baseline that involves no fine-tuning that allows for interpretable results (e.g. via visualisation of nearest prototypes in a cluster for a given query). The approach involves extracting the pre-trained features of a new dataset, performing unsupervised clustering and then post-processing the clustering results to make predictions, in different ways depending on the type of task. The experiments use SWAG-ViT models for the pre-trained representations. It is shown that on some coarse-grained tasks (e.g. CIFAR-10 and CIFAR-100) their unsupervised method is competitive with a purpose-trained supervised ResNet, but weaker on fine-grained tasks. Results are also shown for few-shot learning for the same tasks, showing improved performance. Results are also shown for a continual learning setup, where it is shown to outperform one gradient-based continual learning method (itaml) on incremental CIFAR tasks, and also an unsupervised domain adaptation setup, where it is shown to outperform the method from Peng 2019 on DomainNet. Finally, some examples of retrieved prototypes are visualised in the context of the unsupervised domain adaptation experiments, that demonstrate some failure modes of the method.

**Strengths:**

- The method is simple in the sense that it does not require any fine-tuning of the pre-trained model.
- The method is evaluated in a wide variety of settings, demonstrating its applicability.
- It is an interesting result that the method performs better on coarse-grained tasks that fine-grained tasks, and is informative about the information content of pre-trained representations.
- As demonstrated, the method is amenable to interpretability and has use as a way of evaluating representations of pre-trained models.
- The paper is clear that it is proposing a relevant baseline - it is not trying to claim that the method is clearly superior to other more sophisticated methods. The value is in the simplicity.
- In the same vein, the paper is open about the limitations of the methods and demonstrates several failures in the interpretability analysis.

**Weaknesses:**

- The paper is often confusing to follow and the presentation is quite unclear in places. This makes it difficult for the reader to understand exactly how the method works and what is being shown in the results. E.g.
    - In 3.1.1. “weakly connected component identifiers” are not properly defined, nor how the WeaklyConnectedComponents function works. This seems to be key to the clustering algorithm.
    - Table 1 is hard to read, with the train, valid and test results concatenated together
    - Figure 4 is very hard to interpret as the y axes have different limits and scales for each column in a given row.
    - Boldening the font for the best results in a category would make the tables easier to read.
    - Equation 2 is confusing - how is c^2 defined?
- The literature review and comparison to other methods is somewhat impoverished, particularly for continual learning methods that also make use of pre-trained models. E.g.
    - J. Gallardo, T. L. Hayes, and C. Kanan. Self-supervised training enhances online continual learning. BMVC, 2021. Shows that more general SSL pretraining led to less forgetting on downstream tasks.
    - Lesort, Timothée, Thomas George, and Irina Rish. "Continual learning in deep networks: an analysis of the last layer." arXiv preprint arXiv:2106.01834 (2021). Analyses continual learning using a fixed feature extractor (last layer of a Resnet pretrained on imagenet) and only training a linear layer on top, also investigating non-gradient based approaches based on KNN.
    - A. Chrysakis and M.-F. Moens. Online continual learning from imbalanced data. In International Conference on Machine Learning, pages 1952–1961. PMLR, 2020. Also trains models in a continual learning setup on top of pretrained representations.
    - S. V. Mehta, D. Patil, S. Chandar, and E. Strubell. An empirical investigation of the role of pre-training in lifelong learning, 2022. Shows that pretraining in computer vision helps prevent catastrophic forgetting in continual learning.
    - Shanahan, Murray, Christos Kaplanis, and Jovana Mitrović. "Encoders and ensembles for task-free continual learning." arXiv preprint arXiv:2105.13327 (2021). A method for continual that learns ensembles based on pre-trained visual representations.

**Questions:**

- Is it really fair to compare the proposed method to the one in Peng 2019 on single source unsupervised domain adaptation when using a model that has been pretrained on the whole of Imagenet? Can you quantify how much the source domain actually helps in this experiment vs just running unsupervised learning directly on the target domain and evaluating the clusters with the Hungarian algorithm? Judging from Table 4, the performance on the target domain seems similar for different source domains, apart from quickdraw which seems to transfer badly.
- In 3.1.1. how are “weakly connected component identifiers” defined and how exactly does the WeaklyConnectedComponents function work? This seems to be key to the clustering algorithm.

---

### Official Review · Reviewer_gqXU · 2023-10-30

**Soundness:** 2 fair
**Presentation:** 2 fair
**Contribution:** 1 poor
**Rating:** 3
**Confidence:** 4

**Summary:**

The paper critiques current deep learning methods for their dependence on expensive fine-tuning, inability to handle continual and open-set learning, and lack of interpretability. Instead of focusing on semi- and unsupervised learning through representation learning, the authors suggest analyzing the feature spaces of existing foundational models. Their research demonstrates that by using non-parametric clustering of these latent feature spaces and utilizing pre-trained classifiers on large datasets, many challenges can be addressed without the need for fine-tuning. The authors propose leveraging the latent spaces of classifiers, like SWAG-ViT, that have been trained on vast datasets. This approach can manage continuously evolving data streams in both unsupervised and semi-supervised contexts. Moreover, it offers interpretability by using prototypes. The authors provide a set of experiments in various settings to showcase the performance of the method.

**Strengths:**

- The primary aim of the paper, which seeks to transcend traditional fine-tuning techniques, is commendable and addresses a pertinent challenge in the field.

- The authors have presented a comprehensive suite of experiments, showcasing a thorough exploration of the proposed methodology across various scenarios (with some caveats, see below).

- The methodological description stands out for its clarity, making it accessible for readers and potentially aiding in replication and further research.

**Weaknesses:**

- The core concept of the paper, which centers on the clustering of latent space vectors, is not novel. This technique has been explored and refined in various forms over recent years, as evidenced by key works like Caron et al. (2018), Ji et al. (2019), and Caron et al. (2021). The authors appear to overlook these seminal papers, and the distinctions between their approach and the methods in these references seem subtle at best.

- The paper covers a broad spectrum of topics, which, while ambitious, seems to dilute its core contribution to a specific application. While it showcases results across multiple settings, the outcomes appear to lack a distinct scenario where the method demonstrates superior performance. The related work section and the experimental section, would benefit from .

- The baselines used in the domain-specific experiments appears to be less than ideal. If one of the main claims is that the proposed method is competitive with fine-tuning, then it is necessary to substantiate this claim empirically. In the few-shot learning segment, a comparison with SOTA fine-tuning techniques (like Bit, Kolesnikov et al. 2020) is needed. Additionally, it would be pertinent to compare with clustering techniques, specifically those presented in Caron et al. 2018 and Ji et al. (2019). The continual learning section should also benchmark against robust fine-tuning baselines, such as the one recently presented in Panos et al. (2023).

References
-----------

Caron, M., Bojanowski, P., Joulin, A., & Douze, M. (2018). Deep clustering for unsupervised learning of visual features. In Proceedings of the European conference on computer vision (ECCV) (pp. 132-149).

Caron, M., Touvron, H., Misra, I., Jégou, H., Mairal, J., Bojanowski, P., & Joulin, A. (2021). Emerging properties in self-supervised vision transformers. In Proceedings of the IEEE/CVF international conference on computer vision (pp. 9650-9660).

Panos, A., Kobe, Y., Reino, D. O., Aljundi, R., & Turner, R. E. (2023). First Session Adaptation: A Strong Replay-Free Baseline for Class-Incremental Learning. arXiv preprint arXiv:2303.13199.

Ji, X., Henriques, J. F., & Vedaldi, A. (2019). Invariant information clustering for unsupervised image classification and segmentation. In Proceedings of the IEEE/CVF international conference on computer vision (pp. 9865-9874).

Kolesnikov, A., Beyer, L., Zhai, X., Puigcerver, J., Yung, J., Gelly, S., & Houlsby, N. (2020). Big transfer (bit): General visual representation learning. In Computer Vision–ECCV 2020: 16th European Conference, Glasgow, UK, August 23–28, 2020, Proceedings, Part V 16 (pp. 491-507). Springer International Publishing.

**Questions:**

1) How does the method differ and compare from previous work such as Caron et al. (2018), Ji et al. (2019), and Caron et al. (2021)?

2) How does the method empirically compare against more effective fine-tuning baselines?

3) Did the author tried multiple baselines, pretrained on smaller and larger datasets? It would be interesting to see how the results change based on those factors.

Please refer to the "Weaknesses" section for more details and context on the questions above.

---

### Official Review · Reviewer_zkVN · 2023-11-01

**Soundness:** 2 fair
**Presentation:** 2 fair
**Contribution:** 1 poor
**Rating:** 3
**Confidence:** 4

**Summary:**

This paper does clustering analysis on the latent feature spaces of pre-trained classifiers on large-scale datasets to handle continually evolving data streams in an unsupervised or semi-supervised way. They claim that it can solve a number of problems such as open set recognition, unsupervised and few-shot learning, continual learning and unsupervised domain adaptation, while providing interpretability through prototypes. And the proposed set of evaluations can serve as a simple baseline for quantifying generalisation without fine-tuning of the vision transformer models to a range of downstream tasks. The authors conduct experiments on tasks such as unsupervised, continual and few-shot learning and unsupervised domain adaptation, and demonstrate that, in many cases, the methods achieve competitive generalisation without fine-tuning.

**Strengths:**

The authors do a bunch of experiments on unsupervised offline learning, few-shot active learning, open-set recognition and domain adaptation. And the results show that clustering analysis on the latent features extracted from pre-trained models can yield promising results without finetuning.

**Weaknesses:**

Quality/Clarity: It is not well written and hard to follow. The paper title does not reflect the content. The whole paper talks about the clustering analysis on the latent space, which are helpful to some downstream tasks. And the experiments also show that fine-tuning can improve performance too. For example, Table 1 shows that VIT-H/14 (SWAG+IN1K) with finetuning on ImageNet-1K does better than the corresponding non fine-tuning approach.

Originality/significance: the idea is trivial and I do not see much contribution if it only does clustering on the latent space (which is known and done before)

**Questions:**

1. is it possible to change the title to reflect the content in this paper?
2. Table 1 shows that VIT-H/14 (SWAG+IN1K) with finetuning is better, then what is the motivation for this paper?

---

### Official Review · Reviewer_Zcdi · 2023-11-06

**Soundness:** 1 poor
**Presentation:** 1 poor
**Contribution:** 1 poor
**Rating:** 1
**Confidence:** 4

**Summary:**

This paper compares unsupervised clustering of latent features and finetuning, attempting to claim that clustering is a good-enough solution to replace the existing supervised finetuning approach most practitioners adopt.

**Strengths:**

The idea of revisiting unsupervised methods given rich latent representations from foundation models is quite interesting.

**Weaknesses:**

The writing of the paper leaves much to be desired, specifically the contributions. Ironically the contributions that the authors claim are listed in bullet points in the introduction. As a reader, it is hard to tell what exactly is newly proposed by the authors and what are existing methods or settings. For example, the two algorithms in page 4 would seem to be novel contributions by the authors but according to Section 3.1.1 it seems like these are mere exact descriptions of existing clustering methods.

The experiment section is also quite confusing. There are many diverse tasks listed which might seem like a good thing. However, it is unclear what the exact setting is, what the baselines are, and how these results relate to the main claim. For example, in section 4.2 the author directly jumps into one step of what seems like their method. However, there is no context of the purpose of the experiment and how to understand the results. The figures are even worse which will be elaborated in the questions section.

Overall, the paper is far from the acceptance threshold. The authors should really consider what claims to make and how to support them with theoretical or empirical evidence.

**Questions:**

The authors motivate the need for unsupervised clustering by suggesting the semantic-rich latent space could be utilized for downstream tasks without altering the pretrained model (P1, "The proposed analysis suggests that with a generic enough feature space one could perform continual learning using only shallow learning techniques, such as clustering, within the pre-trained latent spaces without finetuning."). Well, as a matter of fact, people do actually do that. One could easily add an additional classification head (or modules of other purpose) and freeze the pretrained model to serve as a feature extractor. The authors should compare with these lightweight, supervised baselines.

The purpose of the related works section is to contexualize how this work compares to other existing related works. However, the related works section this paper dedicates a small paragraph to "explain" each terminology and provide one or two representative existing works. There is no indication of how these existing works relate to this paper. This exacerbates the difficulty of understanding what exactly are the contributions of this work.

Figures are supposed to help readers understand the results more intuitively. The figures in this work unfortunately don't serve this purpose. For example, Fig 4 shows the results of each individual run separately in a figure (as opposed to presenting the mean with variance) and different methods in different subfigures. This makes it immensely difficult to compare between different methods. Also, it seems like the finetuned variants perform significantly better than the no-finetune clustering counterparts, which does not support the authors claim. Another example is Fig 5 where the retrieve related samples are suppose to serve as prototypes for interpreting model error. The figure is extremely difficult to understand and even after attempting to grasp what is presented, it fails to make the model more interpretable.